# Karyotyping with amniotic fluid in 6,572 pregnant women and pregnancy outcomes——A single-center retrospective study

**Shiyu Wei[1], Yuan Yuan[1], Suhua Tu[2], Chunrong Pang[3], Maomei Chen[3], Min Ren[3]\***

**1** School of Nursing, Southwest Medical University, Luzhou, Sichuan, China, **2** Department of Nursing, Affiliated Hospital of Southwest Medical University, Luzhou, Sichuan, China, **3** Department of Obstetrics, Affiliated Hospital of Southwest Medical University, Luzhou, Sichuan, China

\* renmin0608@163.com

## Abstract

### Aims

To detect abnormal chromosome karyotypes in amniotic fluid cells and to explore the relationship among various prenatal diagnostic indications, karyotypes, and pregnancy outcomes.

### Methods

The data used in this study were obtained from 6,572 pregnant women at high risk for fetal chromosomal abnormalities who visited the Luzhou Prenatal Diagnostic Center for amniocentesis from January 2017 to February 2023. The data were accessed from May to October 2023. Upon admission to the hospital, all pregnant women underwent amniocentesis guided by B-mode ultrasound, followed by karyotyping.

### Results

The culture success rate of amniotic fluid was 99.98% (6,571/6,572), with 216 cases of abnormal karyotypes detected (3.29%), including 3 rare cases. There were significant differences in the detection rates of abnormal karyotypes by indication ($\chi2=449.661$, $P<0.001$), with high-risk noninvasive prenatal testing having the highest rate (36.0%), followed by chromosomal abnormalities in one or both spouses (16.1%). A total of 6,065 cases were followed up (92.3%), and most pregnancy terminations were due to fetal chromosomal abnormalities, specifically numerical abnormalities (86.2%).

### Conclusions

All pregnant women with prenatal diagnostic indications should be actively encouraged to undergo prenatal diagnosis and genetic counseling based on their individual circumstances to provide appropriate reproductive guidance, reduce the risk of abnormal births, and promote eugenics.

**Data availability statement:** We have uploaded the data from this study as supplementary information.

**Funding:** This study was supported by the Luzhou Science and Technology Program Project (2022-SYF-52).

**Competing interests:** The authors have declared that no competing interests exist.

**Abbreviations: CMA**: Chromosome Microarray Analysis; **WES**: Whole Exome Sequencing; **NIPT**: Noninvasive Prenatal Testing; **SCA**: Sex Chromosome Aneuploidy; **KS**: Klinefelter Syndrome; **CNVs**: Copy Number Variations

## 1. Introduction

The incidence of birth defects among newborns in China has been on the rise. The complete liberalization of China's birth policy has provided an opportunity for elderly women to have second or third children. However, environmental pollution and genetic mutations have been shown to increase the risk of birth defects. Birth defects—defined as structural, functional, or metabolic abnormalities arising during embryogenesis—may lead to spontaneous abortion, stillbirth, or congenital malformations. Genetic factors account for approximately 25% of all birth defects, including monogenic and chromosomal defects [1,2]. Studies have indicated that the prevalence of birth defects in China is approximately 5.6%, with chromosomal abnormalities making up a significant proportion [3]. Birth defects not only impact the future quality of life of affected children but also place a significant emotional and financial burden on their families [2,4].

Prenatal diagnosis for high-risk pregnant women is a key measure for secondary prevention of birth defects. Karyotyping is widely recognized as the "gold standard" approach for diagnosing fetal chromosomal abnormalities. Chromosome microarray analysis (CMA) and whole exome sequencing (WES) are the latest technological advancements for the prenatal diagnosis. Research indicates that combining karyotyping with CMA and WES enhances the accurate diagnosis of complex chromosomal abnormalities [5–7]. Currently, it is recommended to use a combination of multiple prenatal diagnostic strategies to achieve accurate results [8]. However, in China, the implementation of prenatal diagnostic technologies varies significantly among hospitals, influenced by factors such as policy, technology, and professional training. Currently, few hospitals in economically developed areas possess the advanced technologies like CMA and WES, and many institutions still rely on traditional amniocentesis and routine karyotype analysis. Our prenatal diagnostic center is located in the relatively underdeveloped southwest region of China, where these traditional testing methods are more cost-effective and are widely accepted by patients. During amniocentesis, karyotype analysis is the most fundamental approach that is widely used for prenatal diagnosis. This method can identify chromosomal abnormalities in the fetus, which helps the mother to decide whether to continue or terminate the pregnancy, thereby reducing the incidence of birth defects [9].

In this study, we collected data on fetal karyotypes from 6,572 high-risk pregnant women who underwent amniocentesis to compare the rate of chromosomal abnormalities and the types of abnormal karyotypes identified by different prenatal diagnostic indications. Our aim was to determine the relationships among these indications, abnormal karyotypes, and pregnancy outcomes. Additionally, the study analyzed the live birth rates and termination rates of fetuses based on different indications for prenatal diagnosis and types of chromosomal abnormalities, to identify factors influencing the pregnancy choices of pregnant women and their families. The findings of this study are expected to further increase awareness about chromosomal karyotype analysis and prenatal diagnosis, as well as increase the rate of genetic counseling, ultimately contributing to the prevention of birth defects.

## 2. Materials and methods

### 2.1 Ethical declarations

This study was approved by the Ethics Review Committee of the Affiliated Hospital of Southwest Medical University (approval code: KY2023141). The general information was collected from pregnant women during genetic counseling after obtaining written informed consent (as outlined in the PLOS consent form) to publish these case details. The collected information, along with the karyotype analysis results, were entered into the Prenatal Diagnosis Information Management System of Sichuan Province, China, for further follow-up.

### 2.2 Study setting and participants

All participants were drawn from the Luzhou Prenatal Diagnostic Center in Sichuan Province, China, a provincial-level prenatal diagnostic sub-center that officially opened in 2017. The staff comprised professionals with experience in performing genetic counseling, testing, ultrasound, and related fields. Data for this study were collected from May to October 2023. A total of 6,572 high-risk pregnant women who underwent amniocentesis at this center from January 2017 to February 2023 were selected, with ages ranging from 18 to 52 years and gestational ages ranging from 16 to 31 weeks. To ensure effective follow-up of study participants, a series of rigorous research designs were implemented. First, prior to study participation, we provided participants with detailed information about the follow-up arrangements and obtained their informed consent. During this process, we communicated with participants, offering information about prenatal screening and diagnosis, helping them understand the study's significance, and establishing a cooperative relationship. Second, we accurately recorded each participant's contact information and developed a comprehensive follow-up plan, specifying the timing, content, and mode of follow-up (e.g., face-to-face interviews, telephone follow-ups, or online questionnaires). Next, we conducted centralized training for all researchers to ensure standardized data collection procedures. Finally, we maintained regular contact with the participants, promptly addressing any questions they had to ensure the smooth progress of the study. The follow-up content was adjusted based on the presence and type of chromosomal abnormalities in the fetus. For families with fetuses that had chromosomal abnormalities, we paid special attention to the child's development. The follow-up visits primarily focused on the specifics of 'Well-Child Care' (Child Health Management), physical development, and any abnormalities in daily behavioral performance.

We defined the inclusion and exclusion criteria and provided uniform training to all members of the research team before the study. The data used in this study were sourced from a well-established database, the Prenatal Diagnosis Information Management System of Sichuan Province. All participants provided written informed consent form before undergoing prenatal diagnoses, and agreed to the use of their amniocentesis data for scientific research. During data collection, multiple contact methods were established to facilitate timely communication with participants when necessary, and our team consistently recorded general information and contact details in the database on the same day. During our follow-up, we implemented the China's unique management model for preventing birth defects, where hospitals and community health service centers collaborate to establish a comprehensive monitoring system. They collaborate to conduct follow-ups with families at high risk for birth defects, aiming to ensure comprehensive data collection. Moreover, the high prioritization of pregnancy-related health information in China motivates individuals to remain engaged and maintain contact throughout the process.

The inclusion criteria were: (1) singleton pregnancy; (2) fulfillment of prenatal diagnostic indications (one indication suffices); (3) karyotyping by amniocentesis; (4) signed informed consent to participate in the study. Exclusion criteria were: (1) recent cell therapy or allogeneic blood transfusion; (2) presence of uterine or pelvic infection; (3) incomplete clinical information.

Indications for prenatal diagnosis included advanced maternal age (actual age or expected delivery date ≥ 35 years), abnormal maternal serum screening tests, congenital anomalies in one or both spouses, chromosomal abnormalities in one or both spouses, pathological ultrasound findings (e.g., polyhydramnios or oligohydramnios, single umbilical artery,

nasal bone hypoplasia, and renal pelvis dilation), high-risk noninvasive prenatal testing (NIPT), abnormal pregnancy history (e.g., unexplained multiple miscarriages, recurrent stillbirths, and children with chromosomal disorders), and history of exposure to teratogenic factors.

## 2.3 Karyotype analysis

Amniocentesis was performed under the guidance of B-mode ultrasound, and 20–30ml of amniotic fluid was extracted, divided into two tubes. It was then centrifuged at 1,800 r.p.m. for 10 minutes, and excess supernatant was discarded, and the remaining fluid was added to two types of culture bottles. Both batches were cultured simultaneously under the conditions of 37°C and 5%$CO_2$. Cell adherence and growth were examined 7 days after inoculation, and the medium was replaced as needed. After 2–3 days, when the cells were actively growing and forming clusters, colchicine was added. The cells were then harvested, prepared, and analyzed by karyotyping.

## 2.4 Statistical analysis

The data were analyzed using IBM SPSS Statistics version 26.0 (IBM Corporation). Count data were expressed as frequencies and constituent ratios. Groups were compared using the chi-squared test or Fisher's exact probability method. All statistical tests were conducted as two-sided tests. Statistical significance level was set at α=0.05, with $P<0.05$ considered statistically significant.

## 3. Results

### 3.1 Distribution of various prenatal diagnostic indications

Among the indications for prenatal diagnosis in the 6,572 pregnant women, advanced maternal age and abnormal maternal serum screening tests accounted for 40.44% (2,658/6,572) and 36.23% (2,381/6,572), respectively, constituting the highest percentages, as shown in **Table 1**.

### 3.2 Distribution of abnormal karyotypes

Of the 6,572 amniotic fluid samples, the culture success rate was 99.98% (6,571/6,572). A total of 216 cases of abnormal karyotypes were detected, representing an abnormality rate of 3.29%. Specifically, numerical abnormalities, structural

Table 1. Distribution of indications for prenatal diagnosis.

| Clinical Indications | Number | Proportion |
|---|---|---|
| Advanced maternal age | 2 381 | 36.23 |
| Abnormal maternal serum screening tests | 2 658 | 40.44 |
| Congenital anomalies in one or both spouses | 24 | 0.37 |
| Chromosomal abnormalities in one or both spouses | 87 | 1.32 |
| Pathological ultrasound finding | 472 | 7.18 |
| High-risk NIPT[1] | 114 | 1.73 |
| Abnormal pregnancy history | 367 | 5.58 |
| History of exposure to teratogenic factors | 116 | 1.77 |
| Family history of genetic disease | 9 | 0.14 |
| Other conditions considered by physicians | 344 | 5.24 |
| Total | 6 572 | 100.00 |

Note:

[1]NIPT: noninvasive prenatal testing

abnormalities, mosaicism, and polymorphisms accounted for 58.3% (126/216), 26.9% (58/216), 6.9% (15/216), and 7.9% (17/216), respectively. Importantly, this study identified rare cases, including trisomy 9, trisomy 9 mosaicism, and trisomy 2 mosaicism. The specific karyotype descriptions are presented in **Table 2**.

### 3.3 Detection of abnormal karyotypes across different prenatal diagnostic indications

Data shown in **Table 3** revealed that the highest detection rate was observed in the high-risk NIPT (41/114, 36.0%), followed by chromosomal abnormalities in one or both spouses (14/87, 16.1%). Autosomal aneuploidy was most frequent among abnormal maternal serum screening tests (40/98, 40.8%), while sex chromosome aneuploidy (SCA) was most common in patients of advanced maternal age (15/28, 53.6%). There were significant differences in the detection rates of abnormal karyotypes across different prenatal diagnostic indications ($P < 0.001$).

### 3.4 Pregnancy outcomes and follow-up

A total of 6,065 participants were followed up, resulting in a follow-up rate of 92.3% (6,065/6,572). **Table 4** presents the follow-up outcomes for pregnancies involving abnormal fetuses, indicating that women of advanced maternal age experienced the highest live birth rate at 32.6%. In contrast, high-risk NIPT pregnancies had the highest termination rate at 23.3%. **Table 5** details the follow-up outcomes for various chromosomal abnormalities. Among live births, structural abnormalities were the most prevalent, accounting for 50.0%. Notably, through a six-month postnatal follow-up, we found that all families with fetal structural abnormalities reported that their child was developing normally. In contrast, terminations were predominantly linked to numerical abnormalities, which accounted for 86.2%. Some of these families reported developmental abnormalities in their children.

## 4. Discussion

In this study, we analyzed 6,571 amniotic fluid samples and identified abnormal karyotypes in 216 cases (3.29%), suggesting that one out of every 30.4 patients who underwent amniocentesis had abnormal karyotypes. This is consistent with the incidence of fetal chromosomal abnormalities in high-risk pregnancies reported previously (2.40%–5.41%) [5,10,11]. However, this prevalence is lower compared with the findings by Younesi et al. [12], who identified abnormal karyotypes in 1,072 out of 14,968 pregnant women, translating to an abnormality rate of 7.2% (one per 13.9). In our study, advanced maternal age and abnormal maternal serum screening tests were the primary prenatal diagnostic indications, which is inconsistent with the findings by Pan et al. [13], who reported that abnormal maternal serum screening tests and pathological ultrasound findings were the top two indications for amniocentesis and cordocentesis. The aforementioned differences in the distribution of prenatal indications, and the increasing proportion of women of advanced maternal age, may be due to the restructuring of China's fertility policy. In response to challenges posed by population aging—particularly regarding quantity, quality, structure, and distribution—China developed the Comprehensive Two-Child Policy in 2015, effectively ending the 35-year-long One-Child Policy. Subsequently, it was revised to a Three-Child Policy in 2021, allowing couples to have three children. These adjustments may have influenced the structure of the childbearing population, increasing the proportion of mothers of advanced maternal age, as suggested by the surveillance data from 2013 to 2017 [14]. Recent advances have made it possible to achieve higher detection rates for CMA and WES. Moreover, the government has supported interventions to promote prenatal screening and diagnosis, which has improved the technical capabilities of hospitals. Our center is progressively adopting new molecular testing technologies, including copy number variation sequencing (CNV-seq). In future, we will leverage these technologies to improve the current understanding in this field.

In this study, autosomal aneuploidy was primarily attributed to trisomy 21 (35.6%), which included 76 homozygous cases and one mosaic case, with a karyotype of XN,+21[10]/46, XN[20]. Trisomy 21, also known as Down syndrome, is the most common chromosomal disorder in newborns, and children with this condition experience developmental challenges after

**Table 2. Detection of chromosomal abnormalities in karyotypes.**

| Types | Name | Karyotype Description | Number (%) | Total Anomaly Rate |
|---|---|---|---|---|
| **Numerical abnormality** | Trisomy 21 | 47, XN, +21 | 76 (35.2) | 1.16 |
| | Trisomy 18 | 47, XN, +18 | 21 (9.7) | 0.32 |
| | Klinefelter syndrome | 47, XXY | 13 (6.0) | 0.20 |
| | Superfemale syndrome | 47, XXX | 3 (1.4) | 0.05 |
| | Supermale syndrome | 47, XYY | 9 (4.2) | 0.14 |
| | Turner syndrome | 45, X | 3 (1.4) | 0.05 |
| | Trisomy 9 | 47, XN, +9 | 1 (0.5) | 0.02 |
| **Structural abnormality** | Robertsonian translocation | 45, XN, rob(13;14)(q10;q10) | 13 (6.0) | 0.20 |
| | | 45, XN, rob(14;21)(q10;q10) | | |
| | | 45, XN, rob(15;21)(q10;q10) | | |
| | | 45, XN, rob(14;22)(q10;q10) | | |
| | Translocation | 46, XN, t(10;22)(q11.2;q11.2) | 24 (11.1) | 0.37 |
| | | 46, XN, t(1;8)(p34;q24.1) | | |
| | | 46, XN, t(5;17)(q13;q23) | | |
| | | 46, XN, t(6;11)(q23;q21) | | |
| | | 46, XN, t(6;7)(q21;p15) | | |
| | | 46, XN, t(6;19)(q21;p13.3) | | |
| | | 46, XN, t(2;5)(q31;p15.3) | | |
| | | 45, XN, t(21;22)(q10;q10) | | |
| | | 46, XY, t(4;17)(q34;q22) | | |
| | | 46, XN, t(4;10)(p15.2q21.2) | | |
| | | 46, XN, t(12;15)(q10;q10) | | |
| | | 46, XN, t(3;10)(p22;q25) | | |
| | | 46, XN, t(2;7)(p23;p22)mat | | |
| | | 46, XN, t(6;17)(p25;q21) | | |
| | | 46, XN, t(4;17)(q34;q22) | | |
| | | 46, XN, t(1;19)(p34;p13.1)pat | | |
| | | 46, XN, t(4;10)(p10;p10)pat | | |
| | | 46,XN,t(8;20)(p24.3;p11.2)mat | | |
| | | 46,XN,t(1;7)(p36.1;p13)mat | | |
| | | 46, XN, t(15;20)(q11.1;q11.1)mat | | |
| | | 46, XN, t(7;12)(p15;q22)pat | | |
| | | 46, XN, t(13;14)(q14;q22) | | |
| | | 46, XN, t(4;21)(p6;q21) | | |
| | | 46, XN, t(6;12)(p12;q12) | | |
| | Deletion | 46, XN, del(10)(p12) | 3 (1.4) | 0.05 |
| | | 46, XN, del(18)(q21.1) [25]/46,XN[25] | | |
| | | 46, XN, del(18) (q21[25]/46,XN[25].1) | | |
| | Inversion | 46, XN, inv(7)(p13q11.1) | 2 (0.9) | 0.03 |
| | | 46, XN, inv(11)(q13.1q23.3)mat | | |
| | Anomaly of sex chromosome | 46, X, del,(X)(q?) | 3 (1.4) | 0.05 |
| | | 46, X, del(X)(p11.2) | | |
| | | 47, XN, del(X)(q26)[10]/46,xx[25] | | |

*(Continued)*

**Table 2.** (Continued)

| Types | Name | Karyotype Description | Number (%) | Total Anomaly Rate |
|---|---|---|---|---|
| | Derivative chromosome | 46, XN, der(14;21)(q10;q10) | 4 (1.9) | 0.06 |
| | | 47, XN, + der(9)t(9;11)(q34;p11.2) pat | | |
| | | 46, XN, der(14;22)(q10;q10) | | |
| | | 46, XN, der(13;14)(q10;q10) | | |
| | Fragments of unknown origin increased | 46, XN, add(14)(q31) [12]/46,XN[15] | 3 (1.4) | 0.05 |
| | | 46, XN, add(13)(p?13) | | |
| | | 46, XN, add(15)(p12) | | |
| | Duplication | 46, XN, dup(18)(q22q23) | 1 (0.5) | 0.02 |
| | Marker chromosome | 47, XN, + mar[3]/46,XN[37] | 5 (2.3) | 0.08 |
| | | 47, XN, + mar | | |
| | | 47, XN, + mar[33]/46,XN[17] | | |
| **Mosaicism** | Trisomy 21 mosaicism | 47, XN, + 21[10]/46, XN[20] | 1 (0.5) | 0.02 |
| | Trisomy 18 mosaicism | 47, XN, + 18[3]/46, XN[20] | 2 (0.9) | 0.03 |
| | | 47, XN, + 18[10]/46, XN[10] | | |
| | Other mosaicism | 47, XN, + 2[4]/46, XN[76] | 2 (0.9) | 0.03 |
| | | 47, XN, + 9[17]/46, XN[13] | | |
| | Turner syndrome mosaicism | 45, X[6]/46, XX[24] | 8 (3.7) | 0.12 |
| | | 45, X[17]/46, XY[31] | | |
| | | 45, X[22]/46, XN[8] | | |
| | | 45, X[14]/46, XN[16] | | |
| | | 45, X[25]/47, XXX[5] | | |
| | | 45, X[19]/47, XXX[11] | | |
| | | 45, X[23]/46, XY[7] | | |
| | | 45, X[10]/46, XN[40] | | |
| | Superfemale syndrome mosaicism | 47, XXX[6]/46, XX[24] | 1 (0.5) | 0.02 |
| | Klinefelter syndrome mosaicism | 47, XXY[10]/46, XY[40] | 1 (0.5) | 0.02 |
| **Polymorphism** | | | 17 (7.9) | 0.26 |

birth [2,15–18]. Advanced maternal age is a major risk factor for trisomy 21, as it is for other chromosomal aneuploidies. Sex chromosome aneuploidy (SCA) was predominantly Klinefelter syndrome (KS) (6.1%), which is the most common SCA and the most prevalent sex chromosome disorder among males, with a prevalence rate of 1 in every 660 newborn males [19]. In our study, 13 cases of homozygosity were detected, one case of mosaicism, with a karyotype of 47, XXY[10]/46, XY[40]. It has been reported that 75% of children with a newborn screening diagnosis of KS experience language delays, and 50% have motor delays [20]. Notably, the prevalence of metabolic syndrome increased from 30.8% to 38.5% during the four years of follow-up among patients with KS, while the incidence of diabetes was 20.5%, significantly higher compared with that of the general population [21]. The risk of thromboembolism and chronic lung disease in males aged 40–70 years with KS is about 3.3 and 4.4 times higher, respectively, compared with normal males [22]. Structural abnormalities were most commonly associated with balanced chromosomal translocations (11.1%). Given that these translocations do not alter chromosome segments, the carriers do not exhibit phenotypic abnormalities and are often diagnosed only after experiencing recurrent miscarriages and reproductive difficulties. Although they present phenotypically normal stature, they may produce abnormal sperm and eggs, which increases the incidence of birth defects. Specific balanced translocations increase the risk

**Table 3. Detection of abnormal karyotypes by prenatal diagnostic indications.**

| Clinical Indications | Number | T21[2] | T18[3] | SCA[4] | T9[5] | SA[6] | Mosaicism | Polymorphism |
|---|---|---|---|---|---|---|---|---|
| Advanced maternal age | 61 | 21 | 7 | 6 | 0 | 18 | 4 | 5 |
| Abnormal maternal serum screening tests | 67 | 28 | 11 | 2 | 1 | 13 | 5 | 7 |
| Chromosomal abnormalities in one or both spouses | 14 | 0 | 0 | 0 | 0 | 14 | 0 | 0 |
| Pathological ultrasound finding | 20 | 8 | 0 | 4 | 0 | 4 | 2 | 2 |
| High-risk NIPT | 41 | 18 | 3 | 15 | 0 | 1 | 4 | 0 |
| Abnormal pregnancy history | 4 | 1 | 0 | 0 | 0 | 3 | 0 | 0 |
| History of exposure to teratogenic factors | 2 | 0 | 0 | 1 | 0 | 1 | 0 | 0 |
| Other conditions considered by physicians | 7 | 0 | 0 | 0 | 0 | 4 | 0 | 3 |
| Total | 216 | 76 | 21 | 28 | 1 | 58 | 15 | 17 |
| χ2 | 449.661 | | | | | | | |
| P | <0.001 | | | | | | | |

Notes:

[2]T21: Trisomy 21;

[3]T18: Trisomy 18;

[4]SCA: sex chromosome aneuploidy;

[5]T9: Trisomy 9;

[6]SA: Structural Abnormalities

**Table 4. Pregnancy outcomes for abnormal fetuses.**

| Clinical Indications | Response (N) | Live Birth (%) | Termination of pregnancy (%) |
|---|---|---|---|
| Advanced maternal age | 59 | 28 (32.6) | 31 (26.7) |
| Abnormal maternal serum screening tests | 65 | 22 (25.6) | 43 (37.1) |
| Chromosomal abnormalities in one or both spouses | 12 | 10 (11.6) | 2 (1.7) |
| Pathological ultrasound finding | 17 | 7 (8.1) | 10 (8.6) |
| High-risk NIPT | 36 | 9 (10.5) | 27 (23.3) |
| Abnormal pregnancy history | 4 | 3 (3.5) | 1 (0.9) |
| History of exposure to teratogenic factors | 2 | 1 (1.2) | 1 (0.9) |
| Other conditions considered by physicians | 7 | 6 (7.0) | 1 (0.9) |
| Total | 202 | 86 | 116 |

**Table 5. Pregnancy outcomes by type of abnormality.**

| Types | Response (N) | Live Birth (%) | Termination of pregnancy (%) |
|---|---|---|---|
| Numerical abnormality | 123 | 23 (26.7) | 100 (86.2) |
| Structural abnormality | 50 | 43 (50.0) | 7 (6.0) |
| Mosaicism | 13 | 5 (5.9) | 8 (6.9) |
| Polymorphism | 16 | 15 (17.4) | 1 (0.9) |
| Total | 202 | 86 | 116 |

of cancer and may be detected as recurrent genetic aberrations in hematological malignancies and solid tumors [23]. The clinical features of these chromosomal abnormalities are continuously being updated.

Besides the commonly reported chromosomal abnormalities, we identified several cases which are less frequently reported. One pregnant woman underwent amniocentesis at 18 weeks due to a high risk of trisomy 18, but prenatal

diagnosis revealed a fetal karyotype of 47, XN, +9. Trisomy 9, first described by Feingold and Atkins in 1973 [24], is a rare aneuploid anomaly associated with severe teratogenic effects and lethality. The characteristic clinical manifestations of trisomy 9 include significant intellectual disability, growth retardation, congenital heart disease, skeletal malformations, and other anomalies. Most studies have focused on trisomy 9 mosaicism [25,26], which was identified in this study. However, complete trisomy 9 is rarely reported in China, with approximately 40 cases documented and fewer than 30 articles in the available literature [27]. A majority of the affected children experience spontaneous abortions in the early stages of pregnancy [28]. Another pregnant woman underwent amniocentesis at 22 weeks due to a high risk of trisomy 21, which was finally detected as a karyotype of 47, XN, +2[4]/46, XN[76]. Studies have demonstrated that when trisomy 2 mosaicism is detected in amniotic fluid cells, it is often associated with structural abnormalities and an increased risk of adverse prognostic outcomes [29–31]. In this study, both cases received adequate genetic counseling, were informed of the potential risks, and ultimately decided to terminate the pregnancies.

In this study, we not only describe the abnormal chromosomal karyotypes detected through karyotyping but also conduct an analysis of the relationship between various prenatal diagnostic indications and abnormal chromosomes, highlighting the strengths of this research. There was a significant difference in the detection rate of abnormal karyotypes by indication ($\chi2=449.661$, $P<0.001$). The detection rate for high-risk NIPT was highest at 36.0%, followed by chromosomal abnormalities in one or both spouses at 16.1%, consistent with the findings by Liu et al. [32]. NIPT is well accepted by pregnant women due to its safety, accuracy, and speed [33,34]. In our study, these women primarily detected abnormal chromosome numbers (87.8%), including 18 cases of trisomy 21, 3 cases of trisomy 18, and 15 cases of SCA. Our findings are consistent with those obtained in previous studies [35,36], confirming the sensitivity and specificity of NIPT in trisomy 21, trisomy 18, and trisomy 13, although its use for SCA screening remains controversial [37–39]. In this study, we found that in cases where one or both spouses had chromosomal abnormalities as a prenatal diagnostic indication, all diagnosed fetal chromosomal abnormalities were structural (100%). Among the fetal chromosomal abnormalities, 8 cases involved one or both spouses being balanced translocation carriers, with 7 cases showing no abnormalities throughout the pregnancy and no special conditions detected in the newborns during follow-up. However, in one case, a fetal derivative chromosome was detected with the karyotype 46, XN, der(9) t(9;11)(q34;p11.2) pat, causing induced abortion.

The presence of fetal structural abnormalities or soft markers on ultrasound should raise a strong suspicion of chromosomal abnormalities. In this study, the detection rate of chromosomal abnormalities based on pathological ultrasound findings was relatively low (4.2%), with numerical abnormalities predominating. Karyotype analysis is limited by its resolution and cannot detect minor copy number variations (CNVs). However, CMA, a high-resolution genomic technology, can accurately identify microdeletions and microduplications at a 50–100 kb level. Studies have shown that for fetuses with abnormal ultrasound findings but a normal karyotype, CMA not only improves the detection rate of pathogenic CNVs but also provides a more comprehensive molecular basis for genetic counseling, guiding pregnancy management, and long-term prognosis assessment [40,41]. Consensus guidelines indicate that fetal structural abnormalities detected via ultrasound are a key indication for CMA, with its application increasing diagnostic yield by 4%-6% [42]. CMA is not routinely available in our hospital due to technical limitations. We must integrate clinical needs with laboratory capacity building to promote the standardized application of high-throughput genomic technologies in prenatal diagnosis.

Karyotype analysis provides relatively reliable results in identifying fetal chromosomal abnormalities, which often determines the pregnancy choices of women and their families. During the follow-up, significant differences in live birth and termination rates were observed across various prenatal diagnostic indications and chromosomal abnormality types. Specifically, advanced maternal age was associated with the highest live birth rate, while abnormal maternal serum screening tests were linked to the highest termination rate. Additionally, the termination rate for chromosomal numerical abnormalities was significantly higher than that for other types. In contrast, many families with fetal chromosomal structural abnormalities chose to continue the pregnancy. Although these newborns developed normally during our follow-up, we cannot rule out the possibility that they may develop phenotypic abnormalities in the future. One of the limitations of our study is the short follow-up period,

necessitating further investigations. Our findings indicate that the types of abnormalities and their possible consequences are significant factors, but there are other factors upon which women and their families make pregnancy decisions. For instance, one fetus was diagnosed as 45, X[14]/46, XN[16], and the mother chose to continue the pregnancy, while in another case, 45, X[6]/46, XX[24], the mother opted for induced labor. Research has shown that pregnancy choices are highly personal and may be influenced by factors such as age, abnormal pregnancy history, gestational age, prior test results, education, values, and income [43–45]. Therefore, when providing genetic counseling to couples, healthcare professionals should fully assess their expectations for the offspring and risk tolerance when making pregnancy decisions.

## 5. Conclusions

This study confirms that karyotype analysis of amniotic fluid cells is a vital tool for prenatal diagnosis. Mid-trimester karyotyping in high-risk pregnant women allows prompt detection of fetal chromosomal abnormalities, particularly in cases of high-risk NIPT and chromosomal abnormalities in one or both spouses. This study also identified several rare chromosomal karyotypes which may offer valuable insights for clinical practice. This study has provided important insights into the pregnancy choices of women with varying characteristics and the factors influencing those choices, serving as a guidance for future genetic counseling.

## 6. Limitations of the study

These findings should be interpreted with several limitations in mind. First, this was an observational study. Although measures were taken to address potential bias, residual or unknown confounding cannot be excluded. Second, all samples were obtained from a single center, which may limit the generalizability of the results. Third, this study lacks information on long-term postnatal follow-up. Therefore, future well-designed multicenter studies with longer follow-up to continue monitoring the offspring and evaluate the data accordingly.

## Supporting information

**S1 File.**
(ZIP)

## Acknowledgments

We want to thank all the pregnant women who participated in our study.

## Author contributions

**Conceptualization:** Min Ren.

**Data curation:** Yuan Yuan.

**Formal analysis:** Yuan Yuan.

**Funding acquisition:** Min Ren.

**Investigation:** Shiyu Wei, Maomei Chen.

**Methodology:** Chunrong Pang.

**Project administration:** Suhua Tu.

**Supervision:** Suhua Tu.

**Writing – original draft:** Shiyu Wei.

**Writing – review & editing:** Min Ren.

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
