## [Decision Letter · Decision Letter 0]

2 Sep 2024

PONE-D-24-30679Karyotyping with amniotic fluid in 6,572 pregnant women and pregnancy outcomes——A single-center retrospective studyPLOS ONE

Dear Dr. Ren,

Thank you for submitting your manuscript to PLOS ONE. After careful consideration, we feel that it has merit but does not fully meet PLOS ONE’s publication criteria as it currently stands. Therefore, we invite you to submit a revised version of the manuscript that addresses the points raised during the review process.

We look forward to receiving your revised manuscript.

Kind regards,

Lingshan Gou, Ph.D.

Academic Editor

PLOS ONE

Journal requirements: 1. When submitting your revision, we need you to address these additional requirements. Please ensure that your manuscript meets PLOS ONE's style requirements, including those for file naming. The PLOS ONE style templates can be found at https://journals.plos.org/plosone/s/file?id=wjVg/PLOSOne_formatting_sample_main_body.pdf and https://journals.plos.org/plosone/s/file?id=ba62/PLOSOne_formatting_sample_title_authors_affiliations.pdf. 2. PLOS requires an ORCID iD for the corresponding author in Editorial Manager on papers submitted after December 6th, 2016. Please ensure that you have an ORCID iD and that it is validated in Editorial Manager. To do this, go to ‘Update my Information’ (in the upper left-hand corner of the main menu), and click on the Fetch/Validate link next to the ORCID field. This will take you to the ORCID site and allow you to create a new iD or authenticate a pre-existing iD in Editorial Manager. 3. Thank you for stating the following financial disclosure:  [Authors: Min Ren, Qingqing Luo, Chunrong Pang, Maomei Chen, Lisha Yang, Yujiao Zhang, Benlan Yin, Kangfen Li, Jie Lu, Wei Li.Funder: Luzhou Science and Technology Program Project (2022-SYF-52).Min Ren: Conceptualization, funding acquisition and writing–review&editing. Chunrong Pang: Methodology. Maomei Chen: Investigation].  Please state what role the funders took in the study.  If the funders had no role, please state: ""The funders had no role in study design, data collection and analysis, decision to publish, or preparation of the manuscript."" If this statement is not correct you must amend it as needed. Please include this amended Role of Funder statement in your cover letter; we will change the online submission form on your behalf. 4. We note that you have indicated that there are restrictions to data sharing for this study. For studies involving human research participant data or other sensitive data, we encourage authors to share de-identified or anonymized data. However, when data cannot be publicly shared for ethical reasons, we allow authors to make their data sets available upon request. For information on unacceptable data access restrictions, please see http://journals.plos.org/plosone/s/data-availability#loc-unacceptable-data-access-restrictions.  Before we proceed with your manuscript, please address the following prompts: a) If there are ethical or legal restrictions on sharing a de-identified data set, please explain them in detail (e.g., data contain potentially identifying or sensitive patient information, data are owned by a third-party organization, etc.) and who has imposed them (e.g., a Research Ethics Committee or Institutional Review Board, etc.). Please also provide contact information for a data access committee, ethics committee, or other institutional body to which data requests may be sent. b) If there are no restrictions, please upload the minimal anonymized data set necessary to replicate your study findings to a stable, public repository and provide us with the relevant URLs, DOIs, or accession numbers. Please see http://www.bmj.com/content/340/bmj.c181.long for guidelines on how to de-identify and prepare clinical data for publication. For a list of recommended repositories, please see https://journals.plos.org/plosone/s/recommended-repositories. You also have the option of uploading the data as Supporting Information files, but we would recommend depositing data directly to a data repository if possible. Please update your Data Availability statement in the submission form accordingly. 5. Your ethics statement should only appear in the Methods section of your manuscript. If your ethics statement is written in any section besides the Methods, please delete it from any other section. 6. Please include your tables as part of your main manuscript and remove the individual files. Please note that supplementary tables (should remain/ be uploaded) as separate ""supporting information"" files". 7. Please include captions for your Supporting Information files at the end of your manuscript, and update any in-text citations to match accordingly. Please see our Supporting Information guidelines for more information: http://journals.plos.org/plosone/s/supporting-information. 

Reviewers' comments:

Reviewer's Responses to Questions

**Comments to the Author**

1. Is the manuscript technically sound, and do the data support the conclusions?

Reviewer #1: Partly

Reviewer #2: Yes

Reviewer #3: Partly

2. Has the statistical analysis been performed appropriately and rigorously? 

Reviewer #1: Yes

Reviewer #2: Yes

Reviewer #3: I Don't Know

3. Have the authors made all data underlying the findings in their manuscript fully available?

Reviewer #1: No

Reviewer #2: Yes

Reviewer #3: Yes

4. Is the manuscript presented in an intelligible fashion and written in standard English?

Reviewer #1: No

Reviewer #2: Yes

Reviewer #3: No

5. Review Comments to the Author

Reviewer #1: The manuscript covered a large population of more than 6000 pregnant women undergoing amniocentesis, which is the highlight of the whole study. The loss to follow-up rate is minimal as a large cohort. However, the manuscript was not clearly-written, and should be re-write and reviewed by an English native speaker. Moreover, amniotic karyotpye is the classic method for detecting fetal chromosomal anomalies. However, recent advances have made it possible to achieve higher detection rate with CMA or WES. Therefore, this study may not be able to contribute any novel ideas to the academic world.

Reviewer #2: Thank you for such an interesting research, I don't mind voting for acceptance of publishing your article into PLOS ONE journal , just very minor comments:

1- If you please update some of your references.

2- If you please can add a list of abbreviation.

3- What do you mean by: B-ultrasound?

4- The manuscript contains some grammatical errors and awkward phrasing. A thorough proofreading is recommended.

Reviewer #3: ABSTRACT: There are several grammatical errors that need to be corrected to make some sentences understandable.

INTRODUCTION: Several grammatical error and wrong use of tenses. The aim of the study be written clearly at the end of the introduction. Line 82 would better read --- during amniocentesis---.

METHODS: For better understanding, the country where the study was carried out and the cadre of the study centre should be stated. In line 92, the authors should specify where and how they obtained their data, which data they obtained and how the data was recorded. In lines 107 -108, it is not clear what the authors mean by pathological ultrasound finding and abnormal pregnancy history, they should be explain.

RESULTS: Grammatical errors and wrong use of tenses which should be corrected. The sentence in lines 136-138 is not clear, are the authors referring to their work or commenting on the results of others?

DISCUSSION: There are too many grammatical errors and wrong use of tenses with incomplete sentences that make most of this section difficult to understand and follow, the authors may need to get an English language specialist to review the text.

DISCUSSION: In the sentence in lines 163-167, the authors should explain or state the country's fertility policy and how this has led to the increase in the proportions stated. In line 169 what do the authors mean by purity cases. In lines 173-175, the authors should confirm from the reference they have cited (15) as to the most common cause of sex chromosomal aneuploidies, do the mean Kirscher's syndrome or Klinefelter's syndrome?In line 173, what do the authors mean by ---13 cases were purity?

TABLES: Abbreviations should written as footnotes below the tables

6. PLOS authors have the option to publish the peer review history of their article (what does this mean? ). If published, this will include your full peer review and any attached files.

**Do you want your identity to be public for this peer review?** For information about this choice, including consent withdrawal, please see our Privacy Policy .

Reviewer #1: No

Reviewer #2: No

Reviewer #3: No

---

## [Author Response · Author response to Decision Letter 1]

25 Oct 2024

Response to Reviewer#1:

(1)Comment 1: The manuscript covered a large population of more than 6000 pregnant women undergoing amniocentesis, which is the highlight of the whole study. The loss to follow-up rate is minimal as a large cohort.

Response 1: Thank you for your valuable feedback. Regarding the large sample size and low loss to follow-up rate in this study, we will provide additional details in the Materials and Methods section and offer the following responses.

First, the "Quality Assessment Guidelines for Prenatal Diagnosis and Screening Technologies in Sichuan Province, China" stipulate that the actual follow-up rate for prenatal diagnosis and screening should be at least 90%. Second, prior to formally commencing the study, we implemented a rigorous research design, clearly defining inclusion and exclusion criteria, providing uniform training for the research team, and developing strategies to address potential follow-up issues. Third, our data came from a well-established database, the Prenatal Diagnosis Information Management System of Sichuan Province. We obtained informed consent from all participants before performing prenatal diagnoses, allowing the use of amniocentesis data for scientific research. This may be the main reason for the low loss to follow-up rate. Fourth, during data collection, we gathered multiple contact methods to facilitate timely communication with participants when necessary. Fifth, our team consistently recorded general information and contact details in the Prenatal Diagnosis Information Management System on the same day; this aids in subsequent follow-up tracking. Sixth, during our follow-up, we implemented China’s unique management model for preventing birth defects, where hospitals and community health service centers collaborate to establish a comprehensive monitoring system. They work together to conduct follow-ups with families at high risk for birth defects, aiming to ensure comprehensive data collection. Seventh, we employed effective communication strategies during follow-up, including regular contact and providing necessary information to address postpartum recovery and fetal care issues. Finally, the high prioritization of pregnancy-related health information in China motivates individuals to remain engaged and maintain contact throughout the process.

(2)Comment 2: The manuscript was not clearly-written, and should be re-write and reviewed by an English native speaker.

Response 2: Thank you for your valuable feedback. To address this issue, we will have native English speakers rewrite and review the manuscript to enhance its clarity and readability.

(3)Comment 3: Amniotic karyotpye is the classic method for detecting fetal chromosomal anomalies. However, recent advances have made it possible to achieve higher detection rate with CMA or WES.

Response 3: Thank you for your insightful comment. In response to the question you raised, we would like to provide the following reply. In this manuscript, we not only describe abnormal chromosomal karyotypes detected through karyotyping but also conduct a comprehensive examination of the relationship between various prenatal diagnostic indications and abnormal chromosomes. More importantly, our study identifies several cases that have been less frequently documented in the existing literature. Furthermore, through follow-up, we found significant differences in live birth and pregnancy termination rates across various prenatal diagnostic indications and types of chromosomal abnormalities. In light of this, we explored the factors influencing women and their families in making pregnancy choices. We hope this study will enhance awareness of chromosomal karyotype analysis and prenatal diagnosis, serve as a reference for future genetic counseling, and provide valuable information for parents regarding pregnancy management for fetuses with chromosomal abnormalities, ultimately contributing to the prevention of birth defects.

Thank you for your guidance; we will supplement the highlights of the manuscript in the Introduction and Discussion sections. Karyotyping is widely recognized as the "gold standard" for diagnosing fetal chromosomal abnormalities. Chromosome microarray analysis (CMA) and whole exome sequencing (WES) represent the latest advancements in prenatal diagnostic technologies. Research indicates that combining karyotyping with CMA and WES enhances the accurate diagnosis of complex chromosomal abnormalities. Currently, it is recommended to integrate multiple prenatal diagnostic strategies for more accurate results. However, in China, the implementation of prenatal diagnostic technologies varies significantly among hospitals, influenced by factors such as policy, technology, and professional training. Currently, few hospitals in economically developed areas possess advanced technologies like CMA and WES, and many institutions still rely on traditional amniocentesis and routine karyotype analysis. Our prenatal diagnostic center is located in the relatively underdeveloped southwest region of China, where these traditional testing methods are more cost-effective and widely accepted by patients. Therefore, amniotic fluid karyotyping was used in this study, and we hope to further explore its significance in prenatal diagnosis. This research will deepen our understanding of karyotyping and provide additional references for its use, ultimately contributing to the prevention of birth defects. However, in recent years, with the government's increased emphasis on prenatal screening and diagnosis, hospitals have gradually enhanced their technical capabilities. Our center is also progressively adopting new molecular testing technologies, such as copy number variation sequencing (CNV-seq). In the future, we will leverage these technologies for further research.

Response to Reviewer#2:

(1)Comment 1: If you please update some of your references.

Response 1: Thank you for your valuable feedback. We will update the references in the manuscript as you suggested. The revised references can be found in the updated version of the manuscript. If there are any specific references you believe require further revision, please let us know, and we will make the necessary changes.

(2)Comment 2: If you please can add a list of abbreviation.

Response 2: Thank you for your valuable feedback. We will add a list of abbreviations above the references in the manuscript to improve clarity.

(3)Comment 3: What do you mean by: B-ultrasound?

Response 3: Thank you for your question. Ultrasound is used for diagnosing and treating human diseases. Diagnostic ultrasound instruments can be categorized into four types: A, B, C, and F, with type B being the most commonly used. In the manuscript, B-ultrasound refers to B-mode ultrasound, a common imaging technique that clearly displays various cross-sectional images of the organs and their surrounding structures. To improve clarity, we will change 'B-ultrasound' in the manuscript to 'B-mode ultrasound.'

(4)Comment 4: The manuscript contains some grammatical errors and awkward phrasing. A thorough proofreading is recommended.

Response 4: Thank you for your valuable feedback. To address these issues, we will have a native English speaker thoroughly proofread the manuscript and enhance its overall clarity.

Response to Reviewer#3:

(1)Comment 1: ABSTRACT: There are several grammatical errors that need to be corrected to make some sentences understandable.

Response 1: Thank you for your valuable feedback. We will revise the abstract to ensure that the sentences are clear and easily understandable.

(2)Comment 2: INTRODUCTION: Several grammatical error and wrong use of tenses. The aim of the study be written clearly at the end of the introduction. Line 82 would better read --- during amniocentesis---.

Response 2: Thank you for your valuable feedback. We will correct the grammatical errors and tense usage in the manuscript and clearly state the purpose of the study at the end of the Introduction section. Additionally, we will revise Line 82 to read ‘---during amniocentesis---’ for clarity.

(3)Comment 3: METHODS: For better understanding, the country where the study was carried out and the cadre of the study centre should be stated. In Line 92, the authors should specify where and how they obtained their data, which data they obtained and how the data was recorded. In Lines 107-108, it is not clear what the authors mean by pathological ultrasound finding and abnormal pregnancy history, they should be explain.

Response 3: Thank you for your valuable feedback. In response to your comments, we will make changes and additions to the original manuscript where appropriate. We will clarify the location of the study in the Methods section, specifically noting that it was conducted at the Luzhou Prenatal Diagnostic Center in Sichuan Province, China, a provincial-level prenatal diagnostic sub-center that officially opened in 2017. The staff comprises professionals with experience in genetic counseling, testing, ultrasound, and related fields. Regarding Line 92, we will detail the methods and locations used to obtain and record the data. In Lines 107-108, we will provide further explanations for the terms "pathological ultrasound finding" and "abnormal pregnancy history," clarifying that pathological ultrasound findings refer to fetal anomalies detected during prenatal ultrasound examinations, while abnormal pregnancy history encompasses complications such as miscarriages and stillbirths. We appreciate your suggestions for improving the manuscript and believe these changes will enhance its clarity.

(4)Comment 4: Grammatical errors and wrong use of tenses which should be corrected. The sentence in Lines 136-138 is not clear, are the authors referring to their work or commenting on the results of others?

Response 4: Thank you for your valuable feedback. We will thoroughly review the manuscript for grammatical errors and incorrect tense usage with the help of a native English speaker, making the necessary changes. Regarding the sentence in Lines 136-138, we will revise that section to clearly indicate that it is our work reporting these rare cases. We appreciate your suggestions for improvement.

(5)Comment 5: DISCUSSION: There are too many grammatical errors and wrong use of tenses with incomplete sentences that make most of this section difficult to understand and follow, the authors may need to get an English language specialist to review the text.

Response 5: Thank you for your feedback regarding the grammatical errors and tense usage in the Discussion section. We will consult with a native English speaker for a thorough review and will correct the identified issues with their assistance. We appreciate your valuable suggestions.

(6)Comment 6: DISCUSSION: In the sentence in Lines 163-167, the authors should explain or state the country's fertility policy and how this has led to the increase in the proportions stated. In Line 169 what do the authors mean by purity cases. In Lines 173-175, the authors should confirm from the reference they have cited (15) as to the most common cause of sex chromosomal aneuploidies, do the mean Kirscher's syndrome or Klinefelter's syndrome? In line 173, what do the authors mean by ---13 cases were purity?

Response 6: Thank you for your valuable feedback. We will supplement the details of China's fertility policy in the manuscript, explaining its impact on the increasing proportion of advanced maternal age. In Line 169, we will revise the wording to specify that 'purity cases' refers to 'homozygous cases.' The chromosomal karyotypes of trisomy 21 (Down syndrome) can be classified into three types: homozygous, mosaic, and translocation. Homozygous cases account for 95% of the total incidence of trisomy 21, with all somatic cells in these patients containing an extra chromosome. For Lines 173-175, we will confirm from the reference that Klinefelter syndrome is the most common cause of sex chromosome aneuploidy (SCA) and will revise the manuscript accordingly. Additionally, we will refine the phrasing in Line 173 from 'purity' to 'homozygosity.' We appreciate your valuable suggestions, which enhance the clarity of our manuscript.

(7)Comment 7: TABLES: Abbreviations should written as footnotes below the tables.

Response 7: Thank you for your suggestion. We will add footnotes below each table to clarify the abbreviations used.

---

## [Decision Letter · Decision Letter 1]

14 Jan 2025

PONE-D-24-30679R1Karyotyping with amniotic fluid in 6,572 pregnant women and pregnancy outcomes——A single-center retrospective studyPLOS ONE

Dear Dr. Ren,

Thank you for submitting your manuscript to PLOS ONE. After careful consideration, we feel that it has merit but does not fully meet PLOS ONE’s publication criteria as it currently stands. Therefore, we invite you to submit a revised version of the manuscript that addresses the points raised during the review process.

We look forward to receiving your revised manuscript.

Kind regards,

Lingshan Gou, Ph.D.

Academic Editor

PLOS ONE

Journal Requirements:

Reviewers' comments:

Reviewer's Responses to Questions

**Comments to the Author**

1. If the authors have adequately addressed your comments raised in a previous round of review and you feel that this manuscript is now acceptable for publication, you may indicate that here to bypass the “Comments to the Author” section, enter your conflict of interest statement in the “Confidential to Editor” section, and submit your "Accept" recommendation.

Reviewer #2: All comments have been addressed

Reviewer #3: (No Response)

2. Is the manuscript technically sound, and do the data support the conclusions?

Reviewer #2: Yes

Reviewer #3: Yes

3. Has the statistical analysis been performed appropriately and rigorously? 

Reviewer #2: Yes

Reviewer #3: I Don't Know

4. Have the authors made all data underlying the findings in their manuscript fully available?

Reviewer #2: Yes

Reviewer #3: Yes

5. Is the manuscript presented in an intelligible fashion and written in standard English?

Reviewer #2: Yes

Reviewer #3: Yes

6. Review Comments to the Author

Reviewer #2: Thank you for addressing the previous reviewer's comments. Based on the revised version of your manuscript, I would recommend accepting this article for publication, all the best.

Reviewer #3: There are just a few comments

INTRODCUTION: Lines 78-79 is not clear . Lines 87-88 should be referenced/

METHODS: The authors should state their study design. Line 128-129 is not clear. Line 212 should be ---amniocentesis had abnormal karyotypes--.

DICSUSSION: Lines 292-293 is not clear while 301-303 does not seem complete. The statement in lines 306-308 is not clear, are the authors referring to newborns from their study, if so how did they follow up their developmental patterns because this is not mentioned in the results or methods. Line 318 - offspring should be offsprings, and in line 320, confirm should be confirms

7. PLOS authors have the option to publish the peer review history of their article (what does this mean? ). If published, this will include your full peer review and any attached files.

**Do you want your identity to be public for this peer review?** For information about this choice, including consent withdrawal, please see our Privacy Policy .

Reviewer #2: No

Reviewer #3: No

---

## [Author Response · Author response to Decision Letter 2]

18 Feb 2025

We thoroughly checked the references in accordance with the journal’s requirements to ensure completeness and accuracy. We carefully considered the comments and made revisions to the paper. Responses to the second reviewer’s comments are provided separately, with the corresponding page numbers for the modifications noted.

---

## [Decision Letter · Decision Letter 2]

26 Mar 2025

PONE-D-24-30679R2Karyotyping with amniotic fluid in 6,572 pregnant women and pregnancy outcomes——A single-center retrospective studyPLOS ONE

Dear Dr. Ren,

Thank you for submitting your manuscript to PLOS ONE. After careful consideration, we feel that it has merit but does not fully meet PLOS ONE’s publication criteria as it currently stands. Therefore, we invite you to submit a revised version of the manuscript that addresses the points raised during the review process.

We look forward to receiving your revised manuscript.

Kind regards,

Lingshan Gou, Ph.D.

Academic Editor

PLOS ONE

Journal Requirements:

Reviewers' comments:

Reviewer's Responses to Questions

**Comments to the Author**

1. If the authors have adequately addressed your comments raised in a previous round of review and you feel that this manuscript is now acceptable for publication, you may indicate that here to bypass the “Comments to the Author” section, enter your conflict of interest statement in the “Confidential to Editor” section, and submit your "Accept" recommendation.

Reviewer #4: All comments have been addressed

2. Is the manuscript technically sound, and do the data support the conclusions?

Reviewer #4: Yes

3. Has the statistical analysis been performed appropriately and rigorously? 

Reviewer #4: Yes

4. Have the authors made all data underlying the findings in their manuscript fully available?

Reviewer #4: Yes

5. Is the manuscript presented in an intelligible fashion and written in standard English?

Reviewer #4: Yes

6. Review Comments to the Author

Reviewer #4: The study provides valuable insights into prenatal diagnostics in Southwest China, a region underrepresented in existing literature. By focusing on a large cohort from a resource-limited setting, it highlights the continued relevance of traditional karyotyping alongside emerging technologies. The identification of rare chromosomal abnormalities, such as complete trisomy 9, adds novel clinical data to the sparse literature on these conditions.Furthermore, the analysis of pregnancy decisions based on karyotype severity offers practical guidance for genetic counseling, particularly in emphasizing the critical role of numerical abnormalities in termination choices. These findings complement global prenatal diagnostic research while reflecting China’s unique demographic shifts following recent fertility policy reforms.However, innovation may be limited by the research method because karyotyping is a routine approach, without combining CMA (resolution 50-100 kb) or CNV-seq may lead to missed diagnosis of microdeletion / duplication syndrome (e. g., 22q11.2 deletion syndrome).

To reflect the technology development trend and enhance the depth of research�discussion can be added about CMA. CMA could enhance the detection of pathogenic copy number variations (CNVs), particularly for fetuses with ultrasound anomalies but normal karyotypes.

Minor comments

Line 71�"Birth defects are …… other adverse pregnancy outcomes." can be reduced to�"Birth defects—defined as structural, functional, or metabolic abnormalities arising during embryogenesis—may lead to spontaneous abortion, stillbirth, or congenital malformations."

Line 32 In Abstract method after the last "karyotyping" missing end punctuation, should write as "karyotyping." .

Line 301 “conduct a analysis”�correctly expressed as “conduct an analysis”.

Line 341 “offsprings”should write as“offspring”.

7. PLOS authors have the option to publish the peer review history of their article (what does this mean? ). If published, this will include your full peer review and any attached files.

**Do you want your identity to be public for this peer review?** For information about this choice, including consent withdrawal, please see our Privacy Policy .

Reviewer #4: No

---

## [Author Response · Author response to Decision Letter 3]

2 Apr 2025

Thank you for the opportunity to revise our manuscript. We thoroughly checked the references according to the journal‘s requirements to ensure completeness and accuracy. We have carefully considered the reviewer‘s comments and revised the paper.

---

## [Editor Report · Decision Letter 3]

30 Apr 2025

Karyotyping with amniotic fluid in 6,572 pregnant women and pregnancy outcomes——A single-center retrospective study

PONE-D-24-30679R3

Dear Dr. Ren,

We’re pleased to inform you that your manuscript has been judged scientifically suitable for publication and will be formally accepted for publication once it meets all outstanding technical requirements.

Kind regards,

Lingshan Gou, Ph.D.

Academic Editor

PLOS ONE
---

## [Editor Report · Acceptance letter]

PONE-D-24-30679R3

PLOS ONE

Dear Dr. Ren,

I'm pleased to inform you that your manuscript has been deemed suitable for publication in PLOS ONE. Congratulations! Your manuscript is now being handed over to our production team.

Kind regards,

on behalf of

Dr. Lingshan Gou

Academic Editor

PLOS ONE